# Development of a Third Generation Snack of Rice Starch Enriched with Nopal Flour (*Opuntia ficus indica*)

**DOI:** 10.3390/molecules26010054

**Published:** 2020-12-24

**Authors:** Cecilia Anchondo-Trejo, Jaime Alonso Loya-Carrasco, Tomás Galicia-García, Iván Estrada-Moreno, Mónica Mendoza-Duarte, Lilisbet Castellanos-Gallo, Rubén Márquez-Meléndez, Beatriz Portillo-Arroyo, Cesar Soto-Figueroa

**Affiliations:** 1Faculty of Chemical Science, University Campus II, Food Science and Technology Programme, Autonomous University of Chihuahua, Chihuahua, Chih. CP. 31125, Mexico; cecyanchondo31@gmail.com (C.A.-T.); jaimeloyacarrasco@gmail.com (J.A.L.-C.); lilisbetcastellanosgallo@gmail.com (L.C.-G.); rmarmel@gmail.com (R.M.-M.); bportillo@uach.mx (B.P.-A.); csotof@uach.mx (C.S.-F.); 2CONACyT-CIMAV S.C., Miguel de Cervantes 120, Complejo Industrial Chihuahua, Chihuahua, Chih. CP. 31136, Mexico; ivan.estrada@cimav.edu.mx; 3Research Center for Advanced Materials, CIMAV-Chihuahua, Miguel de Cervantes 120, Complejo Industrial Chihuahua, Chihuahua, Chih. CP. 31136, Mexico; monica.mendoza@cimav.edu.mx

**Keywords:** third generation snacks, modified rice starch, nopal flour

## Abstract

This study aimed to obtain a third-generation snack from native rice starch (NS), rice starch modified by extrusion (MS), nopal flour (NF) and xanthan gum (XG). These raw materials were characterized by proximal analysis, pH, particle size distribution, water absorption index (WAI) and water solubility index (WSI), degree of substitution (DS), differential scanning calorimetry (DSC), rheology, Fourier transform infrared spectroscopy (FT-IR) and scanning electron microscopy (SEM). The analysis of the response variables in the nine formulations of the snack: expansion index (EI), apparent density (AD), hardness (H), luminosity (L*) and tendency to green-red (a*), was performed through a composite central design (CCD), the selected formulations were characterized by SEM. Results showed an increase in WAI, 4.69 ± 0.04, and WSI, 12.61 ± 0.10, for MS, higher than NS values due to chemical modification. According to the color analysis the NF obtained a value of 60.73 ± 0.008 in L* and −6.51 ± 0.004 in a* with green tendency. The DS value obtained was 0.09 ± 0.005, being within the FDA’s permissible range for food use. By FTIR analysis, the acetyl group was corroborated. Finally, employing microwave cooking, snacks made from NS with concentrations of NF (5%) and XG (0%) obtained the highest EI value, 4.47, as well the low Dap and D value (0.37 g/cm^3^, 2.25 N, respectively), corroborated by SEM analysis.

## 1. Introduction

In Mexico about 240 billion pesos are spent per year on the purchase of junk food, according to the Ministry of Health, and only 10 billion pesos are spent on the purchase of basic foods. Up to 40 percent of this junk food corresponds to school spending and is consumed by eight out of ten children; this, together with a sedentary lifestyle, affects 85 percent of basic level children. The consumption of junk food is the main factor for obesity, but so is low physical activity, eating habits and culture acquired from the first years of life. According to the Secretary of Health, obesity has become a significant problem in Mexico. Worldwide Mexico occupies the first and second place in obesity in adults and children, respectively; overweight increased by 40 percent in the last 7 years, the same in obesity in children between 5 and 11 years. In Mexico there are 4.5 million children, 42 million adults and 6 million adolescents who are overweight and obese. Mexico is a country with food deficiency where malnutrition and obesity problems coexist, to the extent that an obesogenic environment is developing that, according to the World Health Organization, is due to the conformation of an environment that promotes consumption with a high content of sugar, fat and salt. The abundant publicity for years has encouraged the consumption of junk foods (snacks, sweets and soft drinks), generating 60 billion dollars a year. In school stores, 9400 million pesos a year are sold in products with high fat content, sodium, or sugar [1]. Foods during frying, lose water on the surface, which is replaced by fat, so it must be drained and even dried with absorbent food paper to decrease the fat intake associated with this process. The frying process improves the quality of ingested fats, through the exchange of lipids, the saturated fatty acids pass to the frying bath, as a counterpart, this is of lower quality and is more easily altered.

With degraded oil, the product absorbs more oil that contains harmful substances (vitamin denaturants, lipid oxidation products such as peroxides and free radicals, as well as gastrointestinal irritants). Carcinogenic compounds such as benzopyrene, benzanthracene and dibenzathracene may also appear, formed by cyclization and dehydrogenation from cholesterol, their consumption being directly related to colon, liver and prostate cancer [2]. All these problems originate in poor diet, either by eating foods with inadequate or deficient nutrients.

Given this problem, the search for alternatives is necessary, such as the development of a healthy snack that has health benefits by containing fiber of natural origin, as well as low oil content, with textural characteristics that are acceptable by the consumer.

The purpose of this research was the development of a third-generation snack from native and modified rice starch, nopal flour and xanthan gum. Through a central composite design, those treatments with the highest affinity to the characteristics of commercial products were analyzed and optimized, so obtaining this snack represents a high potential in the development of healthy snacks with benefits for the consumer.

## 2. Results and Discussions

### 2.1. Proximal Analysis

#### 2.1.1. Native Rice Starch and Modified Starch by Extrusion

Table 1 shows the moisture content of starches, native NS (10.01%) and starch modified by extrusion (MS) (10.33%). These results are within the range reported by Morales [3], who evaluated two varieties of rice (Morelos A-92 and Koshihikari), obtaining values of 9.91 and 11.05%. This difference is mainly due to the different amylopectin/amylose ratio that presents the subspecies to which they belong (indica and japan). Olatunde et al. [4] reported that through chemical modification of native starch it is possible to improve functional properties such as stability, resistance to retrogradation and water retention, increasing in some cases parameters related to this, such as moisture content. This trend is closely associated with the results obtained where the starch showed an increase of 14.97% with respect to the native starch. This difference could be due to the inclusion of functional groups that interact with water, increasing its retention and resulting in less retrogradation.

The ashes were obtained as an inorganic residue from the incineration of NS and MS. The values obtained were 0.14% and 0.13%, respectively (Table 1), a similar value before and after modification. Alvarez [5] reported 0.12% and 0.10% in acetylated starch of quinoa and corn, respectively, there being a behavioral relationship with respect to this research. Mello et al. [6] explained that said reduction was due to the washing of starch during and after acetylation that caused a loss of minerals. On the other hand, Cantellano et al. [7] reported 2.71% protein in rice starch resulting higher than those found in this work where a value of 0.60% in NS and 0.45% in MS was obtained. The protein content in the starch could be affected during the isolation method. González and Pérez [6] characterized native and modified starch by proximal analysis, reaching a value of 0.71% and 0.51% in protein, respectively. These researchers report that the difference was due to starch washes during and after acetylation. The tendency to decrease the protein observed in the different works is influenced by the extrusion process that can generate protein breakdown (denaturation), which impacts and decreases its functional properties [8].

In this work, results of 0.32% fat in NS and 0.13% in MS were obtained, so the starch underwent significant changes since the fat content was considerably reduced after the modification. In comparison with the work of Vargas [9], where values of 30% in native potato starch and 0.17% in acetylated potato starch were obtained, presenting the same tendency as in this work. During the extrusion process, the fats undergo an emulsion process due to the pressure to which they are subjected as they are covered by the starches and proteins, causing an encapsulation of the fat, so the traditional determination of the ethereal extract will not show the actual results corresponding to product [8]. The above can explain the reason for the decrease in fat in the starches analyzed. As for fiber, the value obtained was 0%. During the starch isolation process, all the fraction that can contain this component is removed from the cereal grain. This result only confirms that the method for isolating the starch was carried out optimally.

#### 2.1.2. Nopal Flour

Table 1 shows the results obtained in the proximal analysis of the nopal flour that was another indispensable raw material in the elaboration of third generation snacks. The values obtained are consistent with those reported by Astello-García et al. [10] where they determined the proximal analysis of 5 samples of nopal flour that was harvested at different ripening days (60–100 days, d), reporting a humidity range in the nopales of 4.06 and 5.02%. In this same study the percentage of protein was similar (13.10%) to the present work (13.23%). In fat analysis they found a value of 1.96% in the cactus, very close to the 2.12% found in this work. For nopal 200 d, 14.91% of fiber was reported and for nopal 60 d 18.41% ash, both values are higher than those of this research (9.63% and 16.69% respectively). According to the bibliography aforementioned, they found that the moisture content, the fiber content and the ashes increased as a function of the state of maturation with the passage of days, otherwise with the protein and fat content that decreased over time, as well as the variety analyzed, soil conditions, the type of fertilizer used and the level of rainfall since all these factors modify the nutritional value. The increase in ash content is mainly due to the increase in calcium content during ripening.

### 2.2. Physicochemical Analysis of the Raw Material

Table 2 shows the results obtained from the physicochemical analyzes on native rice starch, modified by extrusion and nopal flour.

#### 2.2.1. Absorption Index (WAI) and Water Solubility (WSI)

A WAI of 1.83 ± 0.04 and 0.47 ± 0.02 of WSI was obtained in NS, while in MS a WAI value of 4.69 ± 0.04 and 12.61 ± 0.10 of WSI was determinated, a with a difference of 2.86 (WAI) and 12.14 (WSI) between the corresponding starches. Through the data reported by Abida et al. [11], there is a similarity to the value of WAI in rice starch “Kohsar”. However, they obtained a higher WSI value (2.00 ± 0) compared to the NS, which can be attributed to the variation in the structure of the granules and to the differences in the amylose content. This polymer can diffuse out of the granule and can solubilize in water. As more amylose percentage contains the starch granule, the higher its solubility index. According to the values obtained, MS presents higher indexes values than NS. González and Pérez [6] studied the effect of acetylation on the properties of rice starch and observed that acetylated rice starch had higher WAI and WSI values compared to native starch due to the introduction of acetyl groups in the rice starch. These groups, as they facilitate the access of water to amorphous areas, for NF, a value of 4.70 ± 0.16 and 24.98 ± 0.28 of WAI and WSI were obtained, respectively. It is observed that indexes the values are higher in the NF than in NS and MS. This is mainly due to the soluble fiber in the nopal, which contains mostly mucilage, followed by gums, pectins and hemicellulose, which causes a more significant absorption and water solubility than starches. Chávez-González [12] analyzed the WAI in nopal flour, reporting a value of 5.0, similar to the present investigation. This value is below that reported in flour for tortilla making, which is suitable for snacks since this allows air cells to form during baking giving place to the expansion of the snack. On the other hand, the WSI value is related to the effect of the drying airflow because the solubility increases as the drying airflow decreases [12,13]. This is due to the fact that particle size decreases with an increase in the flow of drying air and the smaller particles are dustier and dissolve more slowly (they have a lower solubility index). So they reported a higher percentage of solubility in the flour of nopal dehydrated by sprinkling compared to other drying methods: in a tunnel, aspersion and fluidized bed. The nopal of variety “Hartón” obtained a higher solubility index compared to the solubility index obtained in this investigation due to the different species analyzed.

#### 2.2.2. pH

A pH of 6.55 ± 0.05 was obtained in NS and 8.70 ± 0.01 for MS. González and Pérez [6] studied the effect of acetylation on the properties of rice starch, obtaining pH values of 6.60 and 5.80 in the NS and MS, respectively, observing a decrease.

#### 2.2.3. Color

The results of the measurements of NS, MS and NF were expressed according to the CIELab system with the rectangular coordinates (L*, a* and b*). The NS had a value of 98.06 ± 0.42 in the luminosity parameter (L*), a value of 0.016 ± 0.013 in a* (green + red trend −) and a value of −0.36 ± 0.052 in b* (yellow trend + to blue −). In contrast, the MS had a result of 97.80 ± 0.38 in L*, −0.06 ± 0.01 in a* and 2.09 ± 0.01 in b*, observing a minimum difference in the values obtained between the two starches. Márquez-Gómez [14] reports luminosity values of 97.4 in asylum rice starch, indicating that high values of L* are an indication that the raw material does not have wild seeds that favor a dark coloration, while for parameter b* reports an increase in the value of rice starch acetylated due to conditioning, as well as the thermal and mechanical effect in the extrusion process. The color of the NF, was characterized by the same parameters that were used for starches. A result of 60.73 ± 0.008 in L*, −6.513 ± 0.004 in a* and a value of 28.55 ± 0.02 in b* was obtained, which indicates that the NF is pale green with a high luminosity, characteristic colors of this raw material. Chávez-Gonzalez [12] and Guevara-Arauza [15] determined the color in nopal flour and obtained results similar to those mentioned, where he reported that the color of the flour is not very intense and changes easily with the addition of natural colors or when other products are added to consume like desserts and cookies.

#### 2.2.4. Particle Size

The highest percentage in the particle size distribution is found in sieve greater than 100-mesh in both starches and the NF. The percentage is greater than 95% in NS, 62.98% in MS and 95.23% in NF, being considerably higher than the rest of the fractions. The size distribution achieved in this work allowed maintaining uniformity in the test as it as to be a highly homogeneous sample. Chávez-Gonzalez [12] reported for nopal flour, 68.3% of particles greater than a particle size of 0.150 mm. This is consistent with the present work; this is because products with a high content of dietary fiber are usually located in particle size between 0.150 and 0.430 mm. The particle size of the starch was affected by the chemical modification process. These values indicate that these are very fine flours, which are recommended in the preparation of snacks since this type of flour causes the formation of pores in the dough, which allows an adequate exit of water vapor during baking and favors the expansion.

### 2.3. Characterization of Native Rice Starch and Extrusion Modified Starch

#### 2.3.1. Degree of Substitution (DS)

Once the starch modification was made, the DS was determined, obtaining a value of 0.09 ± 0.005 and 2.47% ± 0.14 in the percentage of acetylation, being within the FDA’s permissible range for food use. Colussi et al. [16] found a percentage of acetylation of 2.58% and a degree of substitution with a value of 0.10 in acetylated rice starch. These values are acceptable because they are within the range allowed by the FDA. The minimum differences that exist between the two investigations are due to the methodology used to modify. According to the bibliography mentioned above, they report that the starch was kept under stirring with distilled water and its pH was adjusted with sodium hydroxide before and after adding acetic anhydride, to end with another pH adjustment up to 4.5 with hydrochloric acid and with washes to starch with ethyl alcohol. The starches were acetylated with different acetic anhydride concentration. It is reported that the higher the concentration of acetic anhydride, the higher the percentage of acetylation and the degree of substitution. González and Pérez [6] achieved acetylation of rice starch with a low level of acetyl group substitution, a value of 0.03 in DS and 0.67 in the percentage of acetyl groups. In this case, a more significant difference was obtained compared to those obtained in this investigation. It can be explained by the source of the starch and its conditions before modification, since the starch is diluted and does not allow the greater substitution of acetyl groups because there is not enough availability of hydroxyls susceptible to this substitution. In a study by Castellanos-Gallo et al. [17], they analyzed the effect of acetylation on rheological properties in rice starch. They found similar value in a degree of substitution (0.05) and percentage of acetylation (1.43) to that obtained in this work and to the study performed by Colussi et al. [16] mentioned above, in which a methodology carried out with similar conditions and concentrations in starch was used.

#### 2.3.2. Fourier Transform Infrared Spectrometry (FTIR)

The FTIR analysis confirms that the addition of acetic anhydride to the acetylation reaction favors the insertion of acetyl groups into the starch molecule. With the help of the Essential FTIR 3.5.83 Spectroscopy Toolbox software, FTIR of NS and MS is shown in Figure 1.

In the FTIR spectrum of NS, the main characteristic groups can be observed by bands through signals at different wavelengths. In the range of 3650–3000 cm^−1^, a wide band is assigned to the vibrations by stretching and bending of the intermolecular and intramolecular hydroxyl (OH) groups. The band attributed to the vibration by stretching of the bond (CH) of the starch anhydroglucose unit is observed in a range of 3000–2850 cm^−1^. In the region between 400 cm^−1^ and 1250 cm^−1^, characteristic bands of starch are observed, some of them are attributed to the vibrations of the skeleton of the glucosidic bonds α-1.4 (927 cm^−1^), deformation CH and C-H_2_ (860 cm^−1^), and C-C stretch (763 cm^−1^). In the spectrum of the modified starch, signals associated with esterification can be observed by the appearance of characteristic bands of the ester group, which is evidence that the modification was carried out. The signals are observed at wavelengths of 1335 cm^−1^ and 1265 cm^−1^, the latter specifically corresponds to the wavelength directly associated with the presence of the ester group. At a wavelength range from 1540 cm^−1^ to 1650 cm^−1^ a new band corresponding to the vibrations due to the doubling of hydroxyl groups is observed with the introduction of acetyl groups confirming that the modification of the starch was carried out. Salcedo-Mendoza et al. [18] observed the same band at a wavelength of 1650 cm^−1^ in acetylated cassava and yam starches. Tupa [19] reported similar data for native rice starch, where they found signals in the ranges of 3700–3000 cm^−1^, 3000–2800 cm^−1^ and 1645 cm^−1^; they also found absorption bands in 923 cm^−1^ wave, 857 cm^−1^ and 763 cm^−1^. In the acetylated starch. They reported the appearance of characteristic bands of the ester group centered on 1376 and 1244 cm^−1^, these data agree with those found in the present work.

#### 2.3.3. Differential Scanning Calorimetry (DSC)

The DSC was used to characterize the physicochemical changes of NS and MS. The values obtained in this test are shown in Table 3, where the temperatures are reported in duplicate for the transition start temperature (To), peak temperature (Tp) and final transition temperature (Tf), as well as the transition enthalpy (ΔH). The gelatinization temperature of the NS was 71.58 °C ± 0.10 at a ΔH of 24.03 mW ± 0.09, both values are higher than those presented in the MS, ΔH (23.78 mW ± 0.18) and Tp (65.92 °C ± 0.0008). In MS, lower values were obtained in the temperatures (To, Tp and Tf) and enthalpy (ΔH) compared to the NS. It is related to the acetyl groups that cause a reduction of the associative forces in the amorphous regions within the starch granules that disorganize the intra and inter molecular hydrogen bonds that are responsible for stabilizing the structure and those that affect the viscosity of such starches [20].

#### 2.3.4. Rheology

Table 4 shows the results obtained from the viscosity analysis by means of rheology of NS and MS. The NS had a maximum viscosity of 79.74 Pa s at a time of 5.96 min and a gelatinization temperature (Tg) of 75 °C. Higher values of Tg are observed in this starch than in the MS (65.85 °C), reaching a maximum viscosity (81.09 Pa.s). One of the reasons why the MS obtained lower values of Tg compared to the NS is because the modification increases the swelling power and, therefore, the viscosity, solubility and stability of the starch. Anuntagool et al. [21] conducted a study to find out the characteristics of modified rice starch. After this modification, they observed that the viscosity increased with the acetyl content in all starches and is a result of the increase in solubility and swelling power after acetylation. The introduction of these acetyl groups facilitates water access to amorphous areas due to structural disorganization. In addition to this modification, the gelatinization temperature decreases, providing stability to the retrogradation and giving an improved paste. MS presents a reduction in the retrogradation (3.35 Pa.s) compared to the NS that have a higher retrogradation (0.36 Pa.s). The viscoamilograms of both starches are shown in Figure 2 and Figure 3. Ferreira-Cardoso et al. [22] observed that a prepared mixture of extruded rice starch and barley bagasse had a viscosity profile similar to rice starch, but with lower values of gelatinization temperature (59.5 °C), maximum viscosity (0.08 Pa.s) and retrogradation viscosity of 1.16 Pa.s. This phenomenon indicates that, although the sample was exposed to drastic heating during extrusion, the process only partially gelatinized the starch, reducing the gelatinization temperature.

#### 2.3.5. Scanning Electron Microscopy (SEM)

The micrographs obtained from each starch are shown in Figure 4a,b. The NS showed in its image the presence of small polyhedral granules with a size in a range of 5.21–5.33 μm. In the MS micrograph, a structural disorder and a rupture of the granule are observed, presenting a completely amorphous structure, product of the gelatinization of the starch polymer. Colussi et al. [16] carried out a study where rice starch was acetylated in an aqueous medium; they performed a SEM analysis in which they found that rice starch and acetylated rice starch showed a small starch granule of polyhedral form (2–5 μm). While by acetylation, it did not generate an aggregation in the starch granules and also did not modify its shape. Ali et al. [23] conducted a study where it was determined by a SEM analysis how the granular structure of different starches was affected after being subjected to a thermomechanical extrusion treatment. In the obtained micrographs, they observed that the main difference is the loss of starch crystallinity, explained by the fragmentation of the amylose and amylopectin chains, but mainly of amylose, which contributes mostly to the formation of crystals, organized and repeated between these polymer chains. In contrast, in this investigation, where a change in the shape of the modified starch granule was observed. It is because it was not only acetylated but also extruded where the shear stresses in the extruder generated a disorder of the granule, causing a structural breakdown of the same.

### 2.4. Characterization of the Expanded Product

Table 5 showed the statistical analyses of snacks with 9 experimental units, evaluating responses as expansion index (EI), apparent density (AD), hardness (H) and color parameters (L* and a*). The regression analysis is presented in Table 6.

#### 2.4.1. Expansion Index (EI)

Expansion is an important parameter in the production of cereal-based extruded snack foods in terms of functional properties and on the acceptability of the final product [24]. Experimental values of snacks made with NS were reported in a range of 0.32 to 4.4 (Table 5). Regression analysis showed that EI was affected by the interaction NF-XG (*p ≤* 0.1) and NF2-XG (*p ≤* 0.05). The prediction model for EI used the coded variables:(1)YEI= 1.47 + 0.71X1 – 1.81X2 –0.73 X1X2 +2.25 X11X2

Through the prediction model used, 94.31% of the total variation (*p ≤* 0.0438) was explained for the EI values (Table 6). The effect of the FG interaction is shown in Figure 5a, where the maximum values of EI (4.47) were presented at a range of NF (5%) and XG (0%), as the XG value increased (8%) and the concentration of NF was maintained (5%) the value of EI decreased EI (0.84) in snacks made with native starch. Table 7 shows the results obtained from the characterization of snacks made from modified starch, indicating the concentrations of fiber (NF) and Xanthan gum (XG) in each of the treatments developed. According to the results, the EI had values between 1.40 and 2.76. Regression analysis determined that EI was affected by NF (*p* ≤ 0.05), XG (*p* ≤ 0.05) and by the NF2-XG interaction (*p* ≤ 0.05). The encoded variables used by the prediction model for EI are shown below:(2)YEI =2.17 + 0.28X1 – 0.59X2 – 0.38 X11 + 0.47 X11X2

With the prediction model used for the EI values (Table 8), 95.48% of the total variation (*p ≤* 0.0313) was explained and according to Figure 5b the effect of NF and XG is shown, where it is observed that the maximum value of EI reached a concentration of 5% (NF) and 0% (XG). The effect of EI increases as the concentration of XG decreases, as in the case of formulation 1 (2.35) and 6 (1.58), which have the same concentration of NF, despite the higher concentration of XG in treatment 6. The results obtained from EI are related to those reported by Korkerd et al. [24] in second-generation snacks made by extrusion from brown rice and soybean (in a range of 2.14 to 4.09), where the EI values decreased as the extrusion temperature decreased or increased during its elaboration, but the value was even lower when there was low humidity. However, as the extrusion humidity increased, the IE also increased, concluding that this behavior was related to the degree of gelatinization of the starch. Guevara-Guerrero et al. [25] studied the effect of rice and corn flour and bran extrudates and evaluating the expansion, observing an increase in this response with 25% rice bran and 30% corn brand, instead when they increased the concentration observed that the expansion decreased. This same effect occurs in the present study; a maximum expansion in NF concentrations (5%). Otherwise, with the highest concentration of NF (10%) EI decreases from 4.47 to 2.97 in NS and 2.76 to 2.29 in MS.

#### 2.4.2. Apparent Density (AD)

The values obtained from snacks made with NS ranged from 0.14 to 0.37 g/cm^3^ (Table 5). Regression analysis showed that AD was mainly affected by the linear variable XG (*p ≤* 0.01) and by the F2-G interaction (*p* ≤ 0.05). The prediction model explained 86.72% of the total variation (*p ≤* 0.0482) for AD values (Table 6) and used the coded variables:(3)YAD = 0.21 + 0.12X2 − 0.10 X11 X2

The effect of NF-XG is shown in Figure 6a, in the response variable AD where the maximum values (0.37 g/cm^3^) at intermediate values of NF (5%) and maximum values of XG (8%). When increasing the value of NF (10%) and decreasing the value of XG (4%), the AD was of lower value (0.18 g/cm^3^) in expanded snacks. In the case of snacks made with MS, the values obtained in AD were in a range of 0.14 to 0.21 g/cm^3^ and are shown in Table 7. According to the regression analysis, the AD was affected by the linear variable. NF (*p* = 0.22). The prediction model explained 77.19% of the total variation (*p ≤* 0.5401) for AD values (Table 8) and used the following coded variables:(4)YAD = 0.16 − 0.025 X1 + 0.015 X2 + 0.018 X11 − 0.012 X22 − 0.001 X11X2 + 0.025 X1 X22

The maximum values (0.21 g/cm^3^) were found at low concentrations of NF (0%) and high concentrations of XG (8%), as concentrations of XG (0%) decreased and with intermediate concentrations of NF (5%) AD decreased (0.14 g/cm^3^) as shown in Figure 6b. Gómez-López et al. [26] carried out a study on third-generation snacks made from corn and chia starch in which apparent density (AD) was determined, obtaining results between 99.19 and 486.06 Kg/m^3^, observing that the lowest AD values correspond to the snacks that had a higher expansion rate. The regression coefficients of the AD model showed significance (*p ≤* 0.1) in the addition of chia in the quadratic terms (β2). The prediction model used explained 88% of the total variation of the apparent density. In the study conducted by Delgado-Nieblas et al. [27] on the elaboration and characterization of third generation snack foods from pumpkin flour, they obtained that extrusion temperature and moisture content were the main factors that influenced the determination of apparent density. The maximum values they obtained were at a pumpkin flour content (8%), at low extrusion temperatures and low moisture contents. They concluded that it could be due to the fact that because these conditions did not achieve the degradation of starch, which could have caused a lower retention of water in the foam and, consequently, a lower expansion. For those treatments where they obtained a higher expansion rate, it occurred in those that had lower values of apparent density.

#### 2.4.3. Hardness (H)

The experimental values of snacks made with NS varied from 0.78 to 2.25 N (Table 5). The regression analysis showed that H was affected by the linear variables of NF (*p ≤* 0.1) and XG (*p ≤* 0.01), by the F-G2 interaction (*p* ≤ 0.1). The prediction model for H used the coded variables:(5)YH = 1.38 −0.34X1 + 0.68X2 + 0.37X1 X22

The prediction model used explained 92.27% of the total variation (*p ≤* 0.0170) for the values of H (Table 6). Figure 7a shows the effect of the variation in the concentration of rubber in snacks, where the minimum value of H (0.78 N) was presented at a concentration of XG (0%). When increasing the value of XG (8%) with NF (5%), the value of H in expanded snacks was higher (2.25 N).

In snacks made with MS, the values for H varied in a range of 6.05 to 13.12 N (Table 7). The regression analysis showed that H was affected by the linear variables NF (*p ≤* 0.5), XG (*p ≤* 0.01) and by the NF-XG2 interaction (*p ≤* 0.1). The prediction model explained 96.39% of the total variation (*p ≤* 0.0226) for the values of H (Table 8). The coded variables used by the prediction model for H are the following:(6)YH = 9.56+ 2.26X1 + 2.62X2 − 2.29X1X22

Figure 7b shows the effect of the variation of the concentration of rubber and fiber in snacks, as the concentration of fiber and rubber increased, the hardness was greater. The maximum values were obtained at high values of NF (10%) and average values of XG (4%), the minimum values were found at low values of XG (0%) and average values of NF (5%). Gómez-López et al. [26] reported N-penetration Force values in a range of 5.29–23.58 in third-generation snacks made from corn and chia starch. The lowest penetration force obtained was for the snack with a chia percentage content of 2.57. The treatment with the maximum penetration force value was for the treatment with chia content of 17.57%. Significance (*p ≤* 0.05) was reported in the linear (B) and quadratic (B2) chia regression coefficients. The prediction model used explained 91% of the total variation of the penetration force.

#### 2.4.4. Brightness (L*)

The experimental values of snacks obtained with NS varied from 75.92 to 91.09 (Table 5). The regression analysis showed that L* was affected by the linear variable of NF (*p ≤* 0.05) and by the variable F2 (*p ≤* 0.1). The prediction model explained 96.53% of the total variation (*p ≤* 0.0213) for the values of L* (Table 6) and used the coded variables:(7)$$YL^{*}\;=\;78.65-7.59X1\;+\;3.82X11$$

Figure 8a shows the effect of NF-XG, in the response variable L* where the maximum value (91.09) was found at minimum values of NF (0%) and intermediate values of XG (4%) and the minimum value of L* (75.92) was presented at a concentration of NF of 10% and XG of 4% and it is observed that as the concentration of NF was increased, the variable of L* decreased its value, due to the fact that the cactus confers intense color characteristics that make the expanded snack give a low brightness.

In the case of snacks made from MS, the values of L* varied in a range of 72.71 to 89.01 (Table 7) and according to the regression analysis, it turned out that L* was mainly affected by the variable of NF (*p ≤* 0.01), for the quadratic variable NF2 (*p ≤* 0.01), for the NF2-XG interaction (*p ≤* 0.1) and for the NF-XG2 interaction (*p ≤* 0.1). The prediction model for snacks made with AM showed 99.27% of the total variation (*p ≤* 0.0213) for the values of the L* parameter (Table 6) and the coded variables used were shown below:(8)$$YL^{*}\;=\;78.38\;-7.93X1\;+\;2.73X11-2.74X11\;\times \;X2\;+2.94\;X1\;X22$$

The effect of the NF-XG intersection for the response variable L* is shown (Figure 8b). The maximum value (89.01) was obtained at concentrations of 0% (NF) and 0% (XG), the minimum value (72.71) at concentrations of NF (10%) and XG (4%). It is observed that the value of L* decreased as the fiber concentration increased and the minimum value obtained by L* was in one of the formulations where the concentration of NF was the maximum.

#### 2.4.5. Parameter a* (Trend from –Green to + Red)

The a* values obtained from snacks made with MS varied from −7.01 to −0.54 (Table 5). Through the regression analysis, it was observed that the value of the parameter a* was affected by the linear variable NF (*p ≤* 0.01). The prediction model used the coded variables:(9)$$Ya^{*}\;=\;-5.77\;-\;2.16X1\;+\;1.7X11$$

Through the prediction model, 82.15% of the total variation (*p ≤* 0.0256) was explained for the values of a* (Table 6). The effect of NF-XG is shown in Figure 9a, where the maximum value of a* (−7.01) was obtained with the intermediate values of NF (5%) and XG (4%) in the snacks once expanded.In snacks made from MS, the values of a* varied in a range of −6.32 to 0.87 (Table 7) and according to the regression analysis the parameter a* was affected by the variable of NF (*p ≤* 0.01), for the linear variable XG (*p ≤* 0.05), for the quadratic variable NF2 (*p ≤* 0.01) and for the interaction NF2-XG (*p ≤* 0.1). The prediction model for snacks made with MS showed 99.84% of the total variation (*p ≤* 0.0048) for the values of a* parameter (Table 8) and the coded variables used are shown below:(10)YL = −5.24 −2.42 X1 + 1.17X2 + 2.41 X11−0.60X11 X2

Figure 9b shows the effect of the NF-XG intersection for the response variable a* in which the maximum value (0.87) was obtained at concentrations of NF of 0% and XG of 8%. The minimum value was found at concentrations of NF (5%) and XG (0%). The color trend decreased to an intermediate fiber concentration, otherwise when the concentration of NF is minimal, the value of a* increased. Delgado-Nieblas et al. [27] studied the effect of the extrusion process on the elaboration of functional snack foods using corn starch, whole-grain yellow cornmeal and pumpkin flour. As a result, they observed that when the level of pumpkin flour increased, the L* values tended to decrease. The lowest value of the L* parameter was 76, because it was displayed at values higher than 12%.

Felix-Medina et al. [28] reported that the values of parameter L* decreased as fiber was added to processed corn and bean products once extruded, the darkening of the product increased, resulting in a browning coloration. The lowest values of L* were presented at minimum levels of humidity, mainly due to greater friction of the material inside the extruder, causing depolymerization of the rubber and starch molecules.

#### 2.4.6. Optimization of the Third-Generation Snack

According to the analysis carried out on the nine formulations of snacks made with native rice starch and with extrusion modified starch, two treatments were selected for each type of starch using an optimization technique with the help of the Design Expert program v.7.0 (Stat-Ease, 2014) [29]. To define the ranges to be used in the optimization, a characterization was made in commercial snacks (VB, M, TS) obtained from local trade, the analysis performed were those performed to treatments (EI, AD, H, L*, a*). Below are the results of the physical tests of these snacks (Table 9). Once the results were obtained and analyzed in graphical mode using the statistical program (Design-Expert v7.0.0), it was concluded that the most similar to commercial snacks are those corresponding to formulations 1 and 7 (NS and MS) since they presented the results closest to the analysis performed during the characterization.

For the selected treatments (1 and 7), a more specific characterization was carried out by scanning electron microscopy (SEM) analysis.

### 2.5. Scanning Electron Microscopy (SEM)

Figure 10a,b show the micrographs obtained from scanning electron microscopy analysis for formulation 1 and Figure 11a,b show formulation 7 of the powder product after expanding. It can be seen that the structure of the starch was modified during the cooking process in the microwave equipment. The product presented a crystallinity and rubberiness after being expanded. Snack pellets of formulations 1 and 7 were analyzed before and after expanding. Figure 12 shows the image of pellets before being expanded from each formulations. It was observed that the structure of the starch granule was modified after the thermal process applied. It generated a granular break in the pelellets a gelatinization and a plasticization corresponding to Formulation 1 (Figure 12a–c). In this formulation Xanthan gum was incorporated into the pellet, unlike Formulation 7 (Figure 12d–f) where the rubber was not incorporated. In the micrographs of the expanded products (Figure 13) a porous structure with thin walls is observed in which air cells formed during the thermal process in the microwave can be seen by the release of water vapor and carbon dioxide. The differences found between both formulations are the number and size of cells formed that were more in formulation 1 (Figure 13a–c), being larger those presented in formulation 7 (Figure 13d–f). Delgado-Nieblas et al. [30] reported through micrographs obtained by scanning electron microscopy analysis, three images corresponding to different snacks. Observing that in a sample of material for making snacks, there were several granules with different sizes and shapes, presenting oval, circular and tetrahedral shapes. This, due to the different materials with which the mixture was prepared. In the other two images that correspond to an unexpanded snack and another already expanded, it was observed that the most important changes that occurred in both extruded products were rupture, gelatinization and plasticization of the starch granules. A porous structure with thin cell walls can be distinguished in the micrograph of the already expanded snack, presenting uniformly sized air cells distributed in the observation area. This structure was probably formed due to the sudden release of water vapor that was in the pellet before microwave expansion. According to a study conducted by Molina-Rosell [31] on the production of bread with different types of hydrocolloids, where they observed that the specific volume of the bread was smaller when Xanthan gum was used. They also observed that the characteristics of the crumb were modified in the presence of hydrocolloids obtaining greater porosity when they were present. They concluded that Xanthan gum and HPMC (Hydroxypropyl Methyl Cellulose) stood out as good gluten enhancers. Encina-Zelada et al. [32], carried out a study where they analyzed the effect on the composition of hydrocolloids in gluten-free bread. They observed that the bread to which Xanthan gum was added presented greater volume than the bread that did not contain this type of gum. The increase in gum resulted in an increase in bread volume compared to standard performance, while the bread achieved the best organoleptic characteristics. They also observed that the hardness was lower when less rubber was added, but the breads with more rubber had the lowest hardness increase during the storage period (72 h of monitoring).

## 3. Materials and Methods

### 3.1. Raw Material

Native rice starch (NS) (Best Ingredients, NUEVO LEON, Mexico), fresh nopal (*Opuntia ficus*-indica) purchased in the local market (Chihuahua, Mexico) and xanthan gum (XG) (Dupont, Mexico) were used. Modified starch (MS) (acetylated starch) was obtained by an extrusion process.

### 3.2. Starch Acetylation

This modification was made based on the methodology described by Ruiz-Saenz et al. [33] using acetic anhydride as an esterifying agent. Five hundred g of native rice starch was adjusted to a humidity of 22% and 5.5 g of acetic anhydride was added. The sample was stored for 12 h. After this time, 50 mL of sodium hydroxide solution (10%) was added until the pH was adjusted in a range of 8.5 to 9. Once the sample was mixed, it was processed in an extrusion set (Extruder CW, Brabender, South Hackensack, NJ, USA) of single screw with 2:1 compression screw and 2.8 mm die. Temperatures were maintained at 60, 80 and 100 °C in zones 1, 2 and 3, respectively. The extrudate was dried in a circulating air oven at 50 °C to a humidity of 11%. Finally, the sample was passed through a knife mill (Retsch SK 100, Hann, Germany) and through a sieve 60 (250 mm). The starch obtained was stored in a cold room (5 °C) until use.

### 3.3. Nopal Conditioning

It was performed according to the methodology [34]. The prickly pear cactus was cleaned and washed to remove impurities and avoid contamination of the product. Subsequently, a chopped was performed where 2 cm thick slices were obtained to facilitate rapid and homogeneous drying. Once the nopales were sliced, they were immersed in boiling water for 3 min (1:3 ratio). The scalloped nopales were placed in perforated trays and were dehydrated (Ecoshel FCD 3000, McAllen, TX, USA) for a time of 4 to 6 h in an oven with air circulation (7.5 m/s) until reaching a humidity of 9% as a control. The grinding of the dehydrated material was carried out in a hammer mill (Pulvex, Mexico City, Mexico) with 0.5 mm mesh, to subsequently pass through a series of sieves (mesh No. 20–100, 840–150 μm).

### 3.4. Technological Process

The NS and MS were mixed in an established proportion through a preliminary stage. NF and XG concentrations of each of the treatments were adjusted according to a central design composed of nine experimental units using the statistical program Design Expert v.7 [29]. Three different concentrations of NF (0%, 5% and 10%), XG (0%, 4% and 8%), and constant ingredients as starch (native and modified) were used.

Each of the experimental units underwent a process for the formation of sheets based on wheat was reported by Penna [35], in which the water was placed in a stainless-steel saucepan and heated to 35 °C, the NS and MS were added and stirred for 15 min. After that time, the NF and XG according to the formulation, were mixed. It was stirred with circular motion for 10 min. The cooked mixture was poured into a non-stick pan and placed in the incubator (Ecoshel FCD 3000, Mc Allen, TX, USA at approximately 50 °C for 6 h, at 4 h the mixture was rotated to facilitate drying. Once the mixture dried, it was cut into sheets of approximately 2 cm by 3 cm. Later, these sheets were placed inside the solar dehydrator (23 Grados S.A, Cuautla, Morelos, Mexico) for 3 h at 70 °C or until it reached a humidity of 12%.

### 3.5. Characterization of Native Rice Starch, Modified by Extrusion and Nopal Flour

#### 3.5.1. Proximal Analysis

The proximal analysis is found in AOAC [36] methodology following the determination method for moisture content (7.003), protein (2.057), crude fiber (7.070), lipids (920.39), ash (14.006) and carbohydrates were calculated by difference, subtracting 100 from the sum of the other components of the proximal analysis [37].

#### 3.5.2. pH

It was performed according to method 02-52.01 [38]. 100 mL of distilled water was heated to boil. Once the water was cold, 10 g of sample was added and it was kept under stirring for 15 s. The pH was measured (Hanna Instruments, Cluj, Romania) and the solution was allowed to stand for 5 min and then stirred for 15 s and the pH was measured. This sequence was repeated 4 times with a total time of 20 min. This test was carried out in triplicate.

#### 3.5.3. Determination of Particle Size

Seventy g of NS and MS were deposited in a sieve stack with aperture numbers: 20, 35, 40, 60, 80 and 100 (850, 500, 425, 250, 180 and 150 μm, respectively) for 10 min in a rotavapor (WS Tyler, model RX-29, Mentor, OH, USA). The result was expressed as a percentage relating the weight of the portion retained on each sieve and the total weight of the sample. This methodology was described by Grajeda-Nieto [39].

#### 3.5.4. Water Absorption Index (WAI) and Water Solubility Index (WSI)

According to the methodology [10], 2.5 g of sample were weighed in 50 mL Falcon tubes and 30 mL of distilled water were added. The tubes were kept under stirring for 30 min at 30 °C; then, the samples were centrifuged at 4000 rpm for 10 min (Centra CL3R, Thermo IEC, Midland, ON, Canada). The weight of the solid residue was recorded and the supernatant was evaporated to constant weight at 105 °C in a laboratory oven with forced air circulation. The WAI was calculated using the weight gain ratio and expressed as grams of water absorbed per grams of dry sample. For WSI analysis, the supernatant was decanted and evaporated at 97 °C in a circulating air oven. The weight of residue was recorded and related to the original sample, the result was expressed as a percentage. Both analysis were performed in triplicate.

#### 3.5.5. Color

Based on ASTM D6290 [40]. A representative sample of each: NS, MS and NF was placed in a Petri dish and five readings (4 ends and one in the center) were made with the help of a colorimeter (Konica Minolta^®^ CR-410, Tokyo, Japan) calibrated with illuminant D65, observer 2°. With the triestimule values recorded L* (luminosity), a* (tendency towards green −, or red +) and b* (tendency towards blue −, or yellow +). This test was performed in triplicate and the result was the average value of the tests.

#### 3.5.6. Degree of Substitution (DS)

It was performed based on the methodology reported by Sulbaran et al. [41], where 1 g of extrusion modified starch was placed in a 250 mL flask and 50 mL of an ethanol-water solution (75% *v*/*v*) was added. The mixture was kept under stirring for 30 min at a temperature of 50 °C. Once the cold mixture was added, 40 mL of KOH at 0.5 N was kept at rest for 72 h, with occasional stirring. After time, each treatment was titrated with a standard solution of 0.5 N HCl, using phenolphthalein as indicator. Simultaneously, a control sample was titrated using native starch. By means of the following relationships, the percentage of acetylation and the degree of substitution were obtained:(11)$$\%\;Acetylation=\frac{\left(\left[mlcontrol-mlsample\right]\times 0.5\times 0.043\times 100\right)}{\mathrm{weight}\;\mathrm{of}\;\mathrm{the}\;\mathrm{sample}\;\mathrm{on}\;\mathrm{dry}\;\mathrm{basis}}$$
(12)$$DS=\frac{162\times \%\;Acetylation}{4300-\left(42\times \%Acetylation\right)}$$

#### 3.5.7. Rheology

It was carried out based on the methodology reported by López-Castejón et al. [42]. Solutions were prepared with the NS and MS samples (2.5 g in 11.5 mL of distilled water); each of the samples were kept under stirring for a time not exceeding 30 min. Subsequently, 1–1.5 mL of the solution was placed in aTA-2000 rheometer equipped with a 40 mm plate in oscillation mode, frequency 1 Hz. Through the software (version 4.5A, TA-Instruments Rheometer, New Castle, DE, USA) the parameters were adjusted to perform the analysis.

#### 3.5.8. Differential Scanning Calorimetry (DSC)

The thermal properties of NS and MS were analyzed using a calorimeter (DSC 4000, Perkin-Elmer, Waltham, MA, USA). The experiment was performed according to the methodology described by Wang et al. [43]. The transition temperature (initial temperature To, peak temperature Tp and final temperature Tc) and the enthalpy of gelatinization (ΔH) were calculated with computer software support (Pyris Software, version 11.1.1.0492, Cleveland, OH, USA).

#### 3.5.9. Fourier Transform Infrared Spectrometry (FTIR)

The characterization of NS and MS by FTIR was performed with a Spectrum Two infrared spectrophotometer (Perkin Elmer Inc.) equipped with a universal ATR module (attenuated total reflectance, attenuated total reflectance) with diamond crystal. For the preparation of the powder sample, a conical punch that requires only a few milligrams was used without any special preparation or starch conditioning. The pressure exerted by the punch on the sample was 100 ± 1 N. The vibrational transition frequencies were reported in wavenumbers (cm^−1^) within the mid-infrared. An average of 34 scans per sample was recorded with a resolution of 4 cm^−1^ in the region of 450 to 4000 cm^−1^. Before each experimental reading, a background scan (without sample) was performed under the same experimental conditions as the sample was analyzed.

#### 3.5.10. Scanning Electron Microscopy (SEM)

The samples of NS and MS were observed in SEM based on the methodology written by Oseguera-Toledo [44]. The samples were placed in a Scanning Electron Microscope (Hitachi SU3500 Scanning Electron Microscope, Tokyo, Japan) to a field of 15 KV to obtain micrographs at 70×, 250×, 2000×, 5000×.

### 3.6. Expanded Product Characterization

#### 3.6.1. Color

Based on ASTM D6290 [40]. A representative sample of each of the formulations was placed in a Petri dish and five readings (4 ends and one in the center) were made with the help of a colorimeter (Konica Minolta^®^ CR-410, Tokyo, Japan), calibrate with illuminant D65, observer 2°. With the registered tristimulus values L* (luminosity), a* (tendency towards green −, or red +). This test was performed in triplicate and the result was the average value of the tests.

#### 3.6.2. Hardness (H)

This analysis was performed on a TA-XT2i texturometer (Texture Techonologies Co., Scarsdale, NY, USA). Strength was measured with punction device, considering three points along the expanded product. The test speed was adjusted to 1 mm/s with distance penetration of 5 mm. The test was carried out on ten samples of each of the formulations. The result for each run was the average of all samples.

#### 3.6.3. Expansion Index (EI) and Apparent Density (AD)

With the help of a digital Vernier (Mitutoyo, Hiroshima, Japan) the dimensions (Length, width and thickness) of 10 samples per formulation were determined before and after being expanded. The specific volume was calculated using the dimensions, and the difference of both volumes was obtained the expansion index. The apparent density was determined using the seed displacement method (millet) according to the technique reported by Penfield and Campbell [45]. In which the volumes displaced by each treatment were related before and after being expanded.

#### 3.6.4. Scanning Electron Microscopy (SEM)

It was carried out according to the methodology reported by Hu et al. [46]. In addition, the analysis was carried out on the sheet and the expanded product of formulations 1 and 7 made from AM. The sheet was placed in the equipment with double-sided tape. The expanded product was first cross-sectioned and then placed in the same way. The samples were observed in a Scanning Electron Microscope (Hitachi SU3500 Scanning Electron Microscope) and thus micrographs were obtained at 70×, 250×, 2000×, 5000×. This analysis was performed on optimized NS and MS samples.

### 3.7. Experimental Design and Results Analysis

The treatments were proposed using an experimental design composed of 9 experimental units at levels of nopal flour concentration (NF) of 0, 5 and 10% and concentration of Xanthan gum (G) of 0, 4 and 8% (Table 10). The response variables analyzed were: expansion index (EI), apparent density (AD), hardness (H) and color (L* and a*), using the statistical package Design Expert v.9.0.3. (Stat-Ease, 2014) [29]. The optimization was carried out in graphic mode based on the values of a commercial snack (TS, M, and VB). The results of the characterization of the raw materials were analyzed using the one-way analysis of variance and the Tukey mean difference test (*p* ≤ 0.05).

## 4. Conclusions

Through the elaboration of third generation snacks, it was found that rice starch is a suitable raw material for the elaboration of this type of products, due to its functional properties. In addition, the starch was chemically modified and mixed with a hydrocolloid, giving better characteristics to the final product. Through the chemical modification of starch, it was observed that the physicochemical properties changed, generating a granular break that allowed a pregelatinization, resulting in an increase in WAI and WSI mainly, which consequently modified their rheology, making the starch have a more viscous consistency. It was possible to characterize the snacks in their physicochemical and structural properties obtaining better results than those currently in the market. According to the expansion of the product, the greatest expansion occurred at intermediate concentrations of fiber (5%) and low concentrations of xanthan gum (0%) in snacks made from native rice starch and extrusion modified starch. Third generation snacks made with extrusion modified starch with formulation 1 (5% NF and 0% XG) and with formulation 7 (5% NF and 0% XG) presented low AD values, so it is concluded that it was due to the addition of MS and XG that conferred the functional properties to the snack, which favored its greater properties to the other formulations.

## Figures and Tables

**Figure 1 molecules-26-00054-f001:**
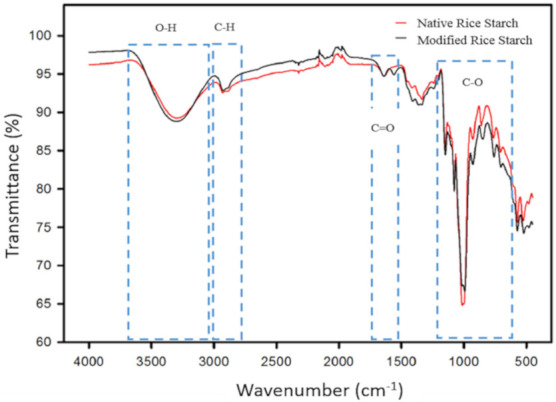
FTIR spectra of native rice starch and extrusion modified rice starch.

**Figure 2 molecules-26-00054-f002:**
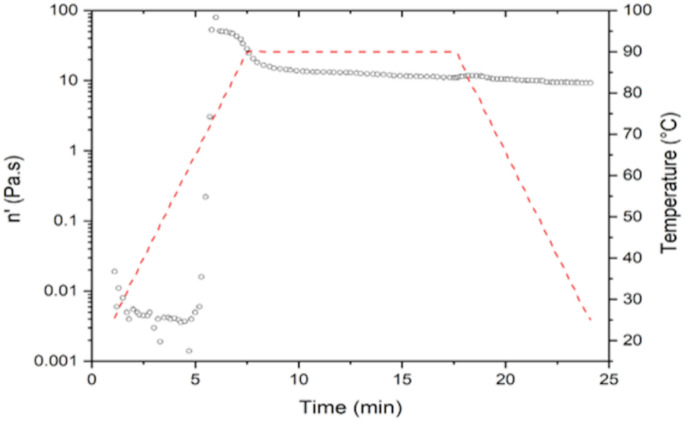
Viscoamilograms of native rice starch.

**Figure 3 molecules-26-00054-f003:**
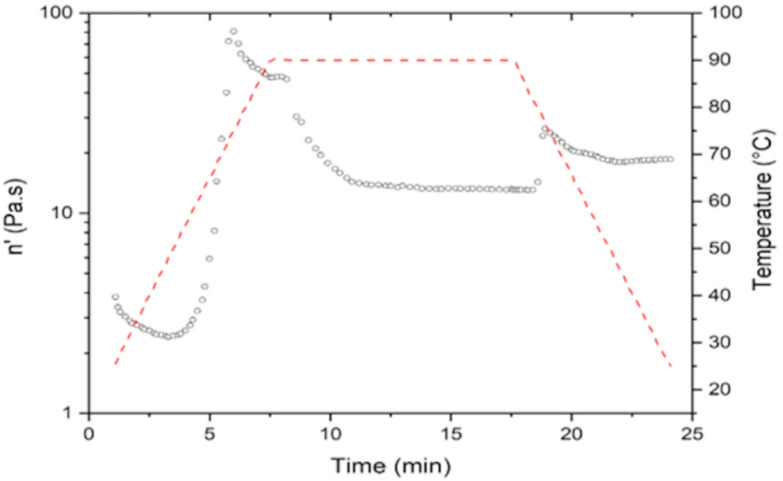
Viscoamilograms of extrusion modified starch.

**Figure 4 molecules-26-00054-f004:**
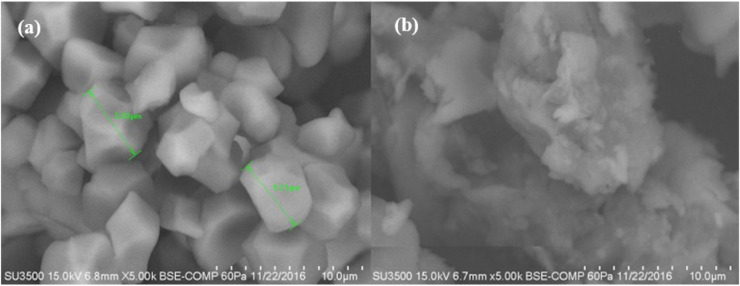
Micrograph of rice starch (**a**) native, and (**b**) modified by extrusion (5000×).

**Figure 5 molecules-26-00054-f005:**
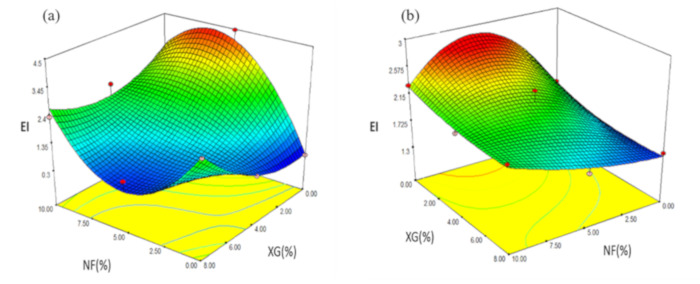
Effect of fiber and rubber interaction in EI in snacks made from (**a**) native rice starch, and (**b**) modified starch.

**Figure 6 molecules-26-00054-f006:**
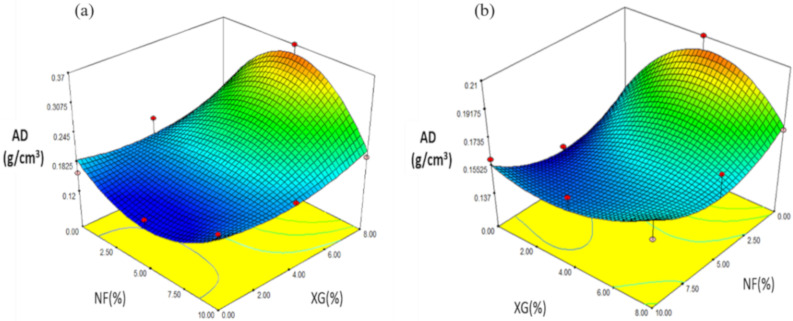
Effect of the interaction of fiber and rubber in the AD (g/cm^3^) of snacks made with: (**a**) native starch, and (**b**) starch modified by extrusion.

**Figure 7 molecules-26-00054-f007:**
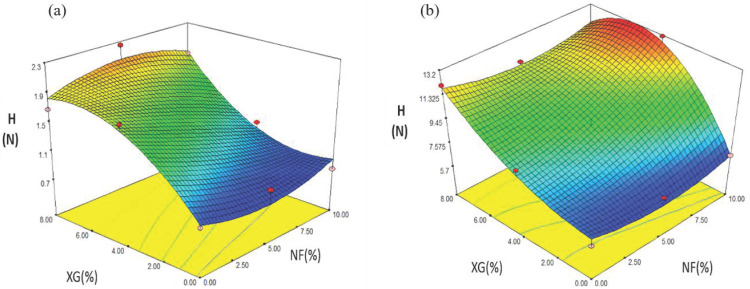
Effect of fiber (NF) and rubber (XG) interaction on the H (N) of snacks made with: (**a**) native starch and (**b**) starch modified by extrusion.

**Figure 8 molecules-26-00054-f008:**
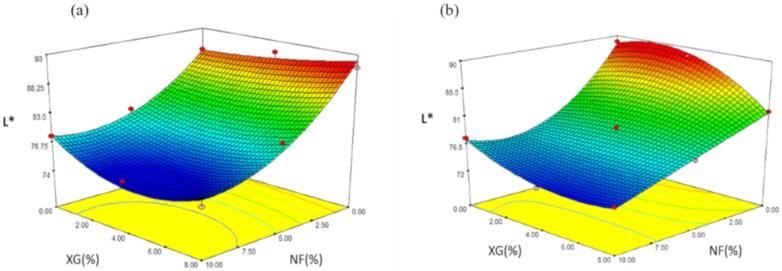
Effect of the interaction of fiber and rubber in the parameter L* in snacks made with: (**a**) native starch and (**b**) starch modified by extrusion.

**Figure 9 molecules-26-00054-f009:**
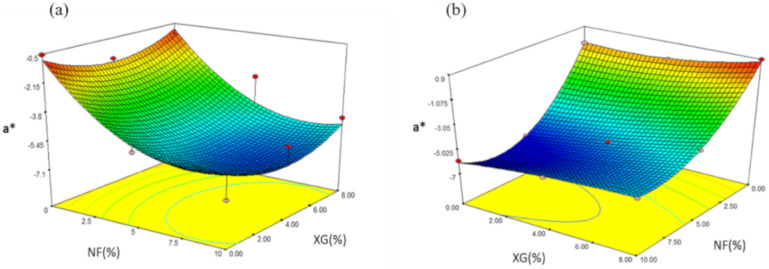
Effect of the interaction of fiber and rubber in parameter a* in snacks made with: (**a**) native starch and (**b**) starch modified by extrusion.

**Figure 10 molecules-26-00054-f010:**
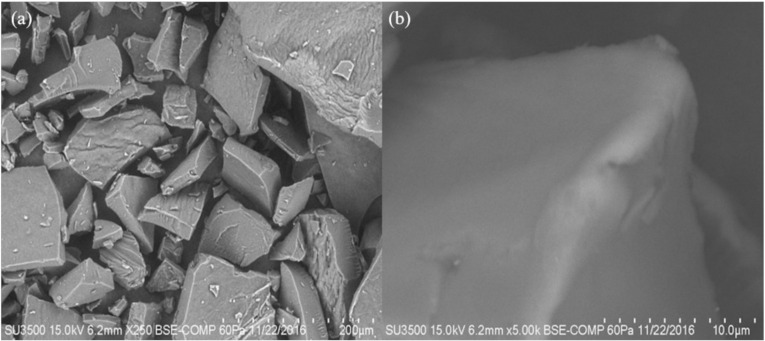
Micrographs of modified starch formulation 1 after undergoing microwave thermal process. (**a**) 250×, (**b**) 5000×.

**Figure 11 molecules-26-00054-f011:**
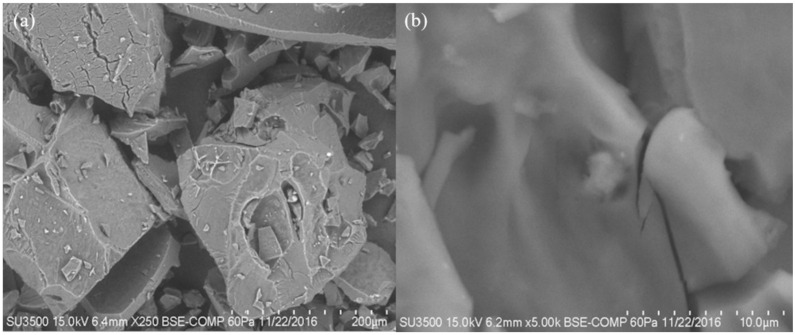
Micrographs of formulation 7 of modified starch after suffering microwave thermal process. (**a**) 250×, (**b**) 5000×.

**Figure 12 molecules-26-00054-f012:**
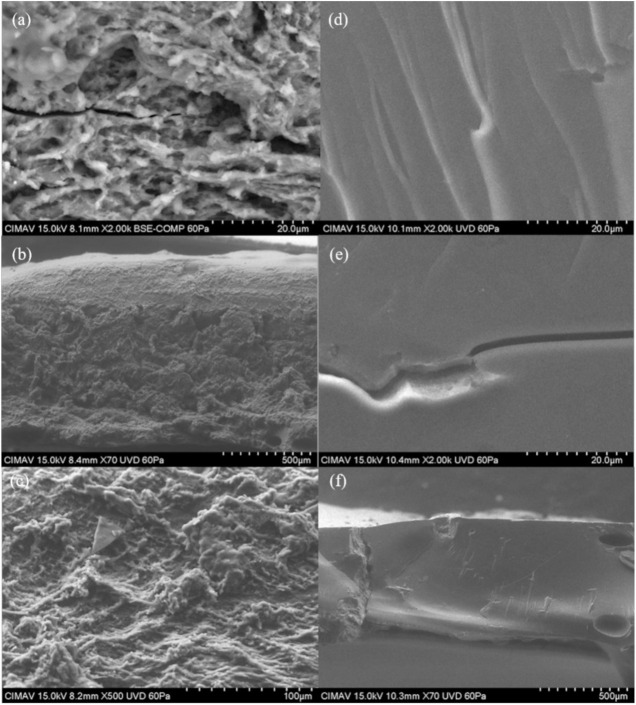
Micrographs of the product without expanding. Plate: (**a**) Formulation 1 (2000×), (**b**) Formulation 1 (70×), (**c**) Formulation 1 (500×), (**d**,**e**) Formulation 7 (2000×), (**f**) Formulation 7 (70×).

**Figure 13 molecules-26-00054-f013:**
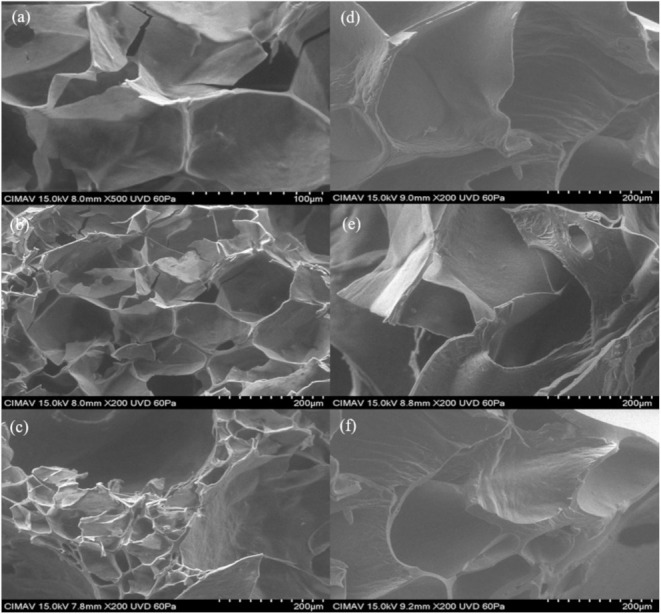
Micrographs of the expanded product. (**a**) Formulation 1 (500×), (**b**) Formulation 1 (200×), (**c**) Formulation 1 (200×), (**d**–**f**) Formulation 7 (200×).

**Table 1 molecules-26-00054-t001:** Results of proximal analysis for native starch, modified starch and nopal flour.

Chemical Properties	NS (%)	MS (%)	NF (%)
Protein	0.60 ± 0.01 ^a^	0.45 ± 0.02 ^b^	13.23 ± 0.58 ^c^
Fat	0.32 ± 0.04 ^a^	0.13 ± 0.00 ^b^	2.12 ± 0.13 ^c^
Ashes	0.14 ± 0.00 ^a^	0.13 ± 0.01 ^a^	16.69 ± 0.12 ^b^
Humidity	10.01 ± 0.04 ^a^	10.33 ± 0.30 ^b^	4.70 ± 0.15 ^c^
Fiber	0	0	9.63 ± 0.04

NS: Native starch; MS: Modified starch; and NF: Nopal flour. The values represent the mean and the standard deviation (±) (*n* = 3). Tukey test, values that do not share letters (a,b,c) for the same row are significantly different *p* < 0.05.

**Table 2 molecules-26-00054-t002:** Physicochemical properties of native rice starch, modified by extrusion and nopal flour.

Physicochemical Property	NS	MS	NF
WAI	1.83 ± 0.04 ^a^	4.69 ± 0.04 ^b^	4.70 ± 0.16 ^b^
WSI	0.47 ± 0.02 ^a^	12.61 ± 0.10 ^b^	24.98 ± 0.28 ^c^
pH	6.55 ± 0.05 ^a^	8.70 ± 0.01 ^b^	4.35 ± 0.07 ^c^
L*	98.06 ± 0.42 ^a^	97.80 ± 0.38 ^b^	60.73 ± 0.008 ^c^
a*	0.016 ± 0.013 ^a^	0.06 ± 0.01 ^a^	−6.51 ± 0.004 ^b^
b*	−0.36 ± 0.052 ^a^	−2.09 ± 0.01 ^b^	28.55 ± 0.02 ^c^

WAI: Water absorption index and WSI: Water solubility index; L*: luminosity; a*: tendency towards green −, or red +; b*: tendency towards blue −, or yellow +. NS: Native starch; MS: Modified starch; and NF: Nopal flour. The values represent the mean and the standard deviation (±) (*n* = 3). Tukey test, values that do not share letters (a,b,c) for the same row are significantly different *p* < 0.05.

**Table 3 molecules-26-00054-t003:** Thermal characteristics of native and extrusion modified rice starch obtained with DSC.

	To (°C)	Tp (°C)	Tf (°C)	ΔH (mW)
NS	65.40 ± 0.09 ^a^	71.58 ± 0.10 ^a^	81.85 ± 0.10 ^a^	24.03 ± 0.09 ^a^
MS	65.56 ± 0.00 ^b^	65.92 ± 0.00 ^b^	66.60 ± 0.00 ^b^	23.78 ± 0.18 ^a^

NS: native rice starch, MS: modified starch by extrusion; Transition temperature (To: initial, Tp: in peak, Tf: final). Tukey test, values that do not share letters (a,b) for the same column are significantly different *p* < 0.05.

**Table 4 molecules-26-00054-t004:** Viscosity characteristics of native and modified extrusion rice starch.

	*n*’ (Pa.s)	Tg (°C)	Retrogradation (Pa.s)
NS	79.745 ± 0.07 ^a^	75 ± 0.33 ^a^	0.36 ± 0.03 ^a^
MS	81.09 ± 0.94 ^a^	65.85 ± 0.20 ^b^	3.35 ± 0.04 ^b^

NS: Native starch; MS: Modified starch; *n*’: viscosity (Pa.s); Tg: gelatinization temperature (°C). Tukey test, values that do not share letters (a,b) for the same column are significantly different *p* < 0.05. The values represent the mean and the standard deviation (±) (*n* = 2).

**Table 5 molecules-26-00054-t005:** Results of the analysis of snacks made from native starch.

Formulation	NF (%)	XG (%)	EI	AD	H	L*	a*
1	5	4	1.08	0.17	1.19	76.58	−7.01
2	0	0	0.32	0.16	0.84	89.22	−0.54
3	10	4	2.61	0.18	1.12	75.92	−5.15
4	0	8	2.65	0.21	1.68	90.97	−1.32
5	10	8	2.38	0.20	1.85	76.00	−4.80
6	5	8	0.84	0.37	2.25	82.15	−3.12
7	5	0	4.47	0.14	0.90	81.35	−5.04
8	0	4	0.73	0.21	1.80	91.09	−1.73
9	10	0	2.97	0.20	0.78	79.88	−6.58

NF: nopal flour; XG: Xhantan gum; EI: Expansion index; AD: Apparent density (g/cm^3^); H: Hardness (N); L*: luminosity; a*: (tendency towards green −, or red +).

**Table 6 molecules-26-00054-t006:** Regression analysis values of snacks made with native starch (Design-Expert 7.0.0).

Coefficients	YEI	YAD	YH	YL*	Ya*
Intercept	1.47	0.21	1.38	78.65	−5.77
Linear					
β1	0.71 **		−0.34 *	−7.59 **	−2.16 ***
β2	−1.81 **	0.12 ***	0.68 ***		
Quadratic					
β11		−0.033 ns		3.82 *	1.7 *
Β22	0.80 ns	0.027 ns		2.06 ns	1.06 ns
Interaction					
β1 * β2	−0.73 *			−1.41ns	
β11 * β2	2.25 **	−0.10 **	−0.20 ns		
β1 * β22			0.37 *	1.51 ns	
R^2^	0.9431	0.8672	0.9227	0.9653	0.8215
*p*≤	0.0438	0.0482	0.0170	0.0213	0.0256

* Level of significance at *p* ≤ 0.1; ** Significance level at *p* ≤ 0.05; *** Level of significance at *p* ≤ 0.01. EI: Expansion index; AD: Apparent density (g/cm^3^); H: Hardness (N); L*: luminosity; a*: (tendency towards green −, or red +); ns: no significance.

**Table 7 molecules-26-00054-t007:** Results of the analysis of snacks made from modified starch.

Formulation	NF	XG (%)	EI	H	AD	L*	a*
1	5	4	2.35	0.14	8.71	79.31	−5.10
2	0	0	1.78	0.15	6.05	89.01	−0.49
3	10	4	2.02	0.16	13.12	72.71	−5.33
4	0	8	1.53	0.17	12.09	82.0	0.87
5	10	8	2.08	0.16	11.43	73.55	−4.98
6	5	8	1.58	0.17	11.21	77.26	−3.97
7	5	0	2.76	0.14	6.38	76.91	−6.32
8	0	4	1.40	0.21	8.6	88.57	−0.48
9	10	0	2.29	0.16	6.58	77.48	−5.90

NF: nopal flour; XG: Xhantan gum; EI: Expansion index; AD: Apparent density (g/cm^3^); H: Hardness (N); L*: luminosity; a*: (tendency towards green −, or red +).

**Table 8 molecules-26-00054-t008:** Regression analysis values of snacks made with modified starch (Design-Expert 7.0.0).

Coefficients	YEI	YAD	YH	YL*	Ya*
Intercept	2.17	0.16	9.56	78.38	−5.24
Linear					
β1	0.28 **	−0.025 ns	2.26 **	−7.93 ***	−2.42 ***
β2	−0.59 **	0.015 ns	2.62 ***		1.17 **
Quadratic					
β11	−0.38 **	0.018 ns	0.88 ns	2.73 *	2.41 ***
Β22	0.083 ns	−0.012 ns	−1.18 ns	−0.83 ns	0.17 ns
Interaction					
β1 * β2				0.77 ns	
β11 *β2	0.47 **	−0.001 ns		−2.74 **	−0.60 *
β1 * β22		0.025 ns	−2.29 *	2.94 *	0.39 ns
R^2^	0.95	0.77	0.96	0.99	0.99
*p≤*	0.0313	0.5401	0.0226	0.0217	0.0048

* Level of significance at *p ≤* 0.1; ** Significance level at *p ≤* 0.05; *** Level of significance at *p ≤* 0.01. EI: Expansion index; AD: Apparent density (g/cm^3^); H: Hardness (N); L*: luminosity; a*: (tendency towards green −, or red +); ns: no significance.

**Table 9 molecules-26-00054-t009:** Results of the analysis of commercial snack.

Snacks	EI	AD	H	L*	a*
TS	2.36 ± 0.26^a^	0.13 ± 0.01 ^a^	4.83 ± 1.26 ^a^	81.33 ± 0.52 ^a^	7.9 ± 0.28 ^a^
M		0.15 ± 0.01 ^a^	5.4 ± 0.65 ^b^	72.27 ± 0.62 ^b^	10.37 ± 0.23 ^b^
VB		0.12 ± 0.006 ^a^	4.63 ± 0.32 ^a^	66.76 ± 0.79 ^c^	9.8 ± 0.48 ^b^

EI: Expansion index; AD: apparent viscosity (g/cm^3^); H: Hardness (N); L*: luminosity; a*: tendency towards green −, or red +. TS, M, and VB: Commercial snacks. Tukey test, values that do not share letters (a,b,c) for the same column are significantly different *p* < 0.05.

**Table 10 molecules-26-00054-t010:** Experimental design.

Formulation	Nopal Flour (NF)	Xanthan Rubber (XG)
1	5	4
2	0	0
3	10	4
4	0	8
5	10	8
6	5	8
7	5	0
8	0	4
9	10	0

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
