# Peer review of "Development of a Third Generation Snack of Rice Starch Enriched with Nopal Flour (Opuntia ficus indica)"

_molecules, 2020, doi:10.3390/molecules26010054_

Round 1
Reviewer 1 Report
The manuscript deals with the development of a third generation snack of rice starch enriched with nopal flour (Opuntia ficus indica).
The English language must be revised.
Please format units in accordance, “50 ºC” not “50 º C”.
Please format units in accordance, “6 h” not “6 H”.
Materials and methods
Line 83- “Subsequently, a chopped or chopped was performed where 2 cm thick slices were obtained to facilitate a fast and homogeneous drying. Once the sliced nopales were immersed in boiling water for 3 min.”???Ratio of water and samples??
Line 85- “The scalloped nopales were placed in perforated trays and were dehydrated (23 Grados S.A, Mexico) for a time of 4 to 6 H in an oven with air circulation until reaching a humidity of 9% as a control.”???air speed (m/s)???
Line 103- “Later these sheets were placed inside the solar dehydrator (23 Grados S.A, Mexico) for 3 H at 70 ° C or until it reached a humidity of 12%.”???why a solar dehydrator???
Line 134- “Based on ASTM D6290 [11]. A representative sample of each samples NS, MS and F was placed in a Petri dish and five readings (4 ends and one in the center) were made with the help of a colorimeter (Miniscan XE 45/0-L, USA). With the triestimule values recorded L * (luminosity), a * (tendency towards green -, or red +) and b* (tendency towards blue -, or yellow +). This test was performed in triplicate and the result was the average value of the tests.”???Illuminant used???ºobserver???
Line 177- “The samples of NS and MS were observed in SEM based on the methodology written by Zazueta-Morales [15]. The samples were placed in a Scanning Electron Microscope (Hitachi SU3500 Scanning Electron Microscope, Japan) and thus obtain micrographs at 70 X, 250 X, 2000 X, 5000 X.”???voltage used???
Line 182- “Based on ASTM D6290 [11]. A representative sample of each of the formulations was placed in a Petri dish and five readings (4 ends and one in the center) were made with the help of a colorimeter (Miniscan XE 45/0-L, USA). With the triestimule values recorded L * (luminosity), a * (tendency towards green -, or red +). This test was performed in triplicate and the result was the average value of the tests.”???Illuminant used???ºobserver???
Line 188- “This analysis was performed on a TA-XT2i texturometer (Texture Techonologies Co, Scarsdale, NY) strength was measured, considering 3 points along the expanded product. The test speed was adjusted to 1 mm/s. The test was carried out on ten samples of each of the formulations. The result for each run was the average of all samples.”???used probe???distance of penetration??
Statistical analysis section???
Results and discussion
This section must be improved. Some subsections present the definition of each parameter and that is not necessary.
Pictures of each sample???
Line 228- “Results of 0.14 % were obtained in rice starch and 0.13 % in rice starch modified by extrusion (Table 2), resulting in a similar value before and after modification.”???Table 2, please add different superscript letters for significant differences. Moreover, please revise the discussion in accordance.
Line 273- “Table 3 shows the results obtained from the physicochemical analyzes performed on the raw material.”???Table 3, please add different superscript letters for significant differences. Moreover, please revise the discussion in accordance.
Line 425- “The values obtained in this test are shown in Table 4, where the temperatures are reported in duplicate of To (Transition start temperature), Tp (transition temperature) and Tf (Final transition temperature), as well as ΔH.”???Table 4, please add different superscript letters for significant differences. Moreover, please revise the discussion in accordance.
Line 449- “Table 5 shows the results obtained from the viscosity analysis by means of rheology of native rice starch and extrusion modified starch.”???Table 5, please add different superscript letters for significant differences. Moreover, please revise the discussion in accordance.
Lines 508 and 548- Tables 7 and 9??Lineal??Cuadratic??Please revise. Moreover, Lack of fit???
Line 682- “To define the ranges to be used in the optimization, a characterization was made in commercial snacks of the brands: "Valentones Barcel", "Mimarca" wheat flour rinds and "Traditional starch" obtained from a local trade, the analyzes performed were those performed to treatments (IE, Dap, Hardness, L *, a *).”??Please codify each brand by letters or numbers.
Line 682- constraints used for optimization???
Line 686- “Once the results were obtained and analyzed in graphical mode by means of the statistical program (Design-Expert v7.0.0), it was concluded that the most similar to commercial snacks are those corresponding to formulations 1 and 7 (AC and AM) since they presented the results (Table 11) closest to the analyzes performed during the characterization.”???Table 10, please add different superscript letters for significant differences. Moreover, Table 11, experimental validation results of the optimized solutions?????how many replicates???please add average value plus standard deviation.
Conclusion
Line 756- “It was possible to characterize the snacks in their physicochemical, structural and sensory properties obtaining better results than those currently in the market. According to the expansion of the product, the greatest expansion occurred at intermediate concentrations of fiber (5%) and at low concentrations of Xanthan gum (0%) in snacks made from native rice starch and extrusion modified starch. Third generation snacks made with extrusion modified starch with formulation 1 (5% F and 0% G) and with formulation 7 (5% FN and 0% G) were the best sensory evaluated, with respect to color, flavor, general acceptability and especially in texture…”??sensory analysis??was it performed??Materials and methods????results???
References
Most of the references (47) have more than 5 years. Please update your list of references.
Please format scientific names in italic, e.g. line 917.
Author Response
Response to Reviewer 1 Comments
The suggested changes to the Manuscript molecules-923965 entitled “Development of a Third Generation Snack of Rice Starch Enriched with Nopal Flour (Opuntia ficus indica)”, are provided by the following means. Each of the changes was made and highlighted in the text with different color based on the suggestion of the reviewer. Thank you for your attention. Best regards.
Dr. Tomas Galicia Garcia
Corresponding Author

Reviewer 2 Report
Manuscript entitled “Development of a Third Generation Snack of Rice Starch Enriched with Nopal flour (Opuntia ficus indica)”. In this paper, the authors investigated the characteristic of native starch, extruded starch and nopal flour, and their mixed products. As presented, the text is not well arranged and there are many mistakes in the detail of the manuscript, and the writer citing too many literatures and fails to combine them with the content very well, so the writing is not acceptable for the journal before the further modification. The following are the questions and some mistakes in this paper:
- In line 25, it should be green-red as the a*, please be more strict. And for CIELab system, there are 3 parameters, so why do you ignore the b* in table 6 and the following analysis.
- In line 62, there is a repetition “with eating food with little” and “lack of intake of them”. In line 84, the words “chopped” appeared twice. Please check the whole passage for this kind of problems.
- Introduction section lack of the specific research content of this paper.
- In the materials and methods section, some indictor should be expressed as formula to make itself clear.
- In line 217 and 257, please confirm it is table 1 that the results were shown in. In line 321, please make sure it is the prickly pear flour that were expressed with method of CIELab. And you need to check the whole passage to avoid the same question.
- Significant difference analysis is missing in allthe tables.
7. In the section of results and discussion, you need focus on your result instead of over describing other people's experiments and conclusions.
Author Response
Response to Reviewer 2 Comments
The suggested changes to the Manuscript molecules-923965 entitled “Development of a Third Generation Snack of Rice Starch Enriched with Nopal Flour (Opuntia ficus indica)”, are provided by the following means. Each of the changes was made and highlighted in the text with different color based on the suggestion of the reviewer. Thank you for your attention. Best regards.
Dr. Tomas Galicia Garcia
Corresponding Author

Round 2
Reviewer 1 Report
The English language must still be revised.
Results and discussion
Line 357- Table 8, lack of fit of the used models???
Line 484- “a characterization was made in commercial snacks of the brands: "Valentones Barcel (VB)", "Mimarca (M) wheat flour rinds and "Traditional snack (TS)”??Please do not use the brands’ names and codify each brand by letters or numbers.
Line 695- which constraints were used for optimization???experimental validation results of the optimized solutions?????how many replicates??
Materials and methods
Line 570- “an oven with air circulation (75 m/s) until”??75 m/s of air speed??not too high??Please check the used air speed.
Conclusion
Line 712- “Third generation snacks made with extrusion modified starch with formulation 1 (5 % NF and 0 % XG) and with formulation 7 (5 % NF and 0 % XG) were the best sensory evaluated, with respect to color, flavor, general acceptability and especially in texture.”???sensory analysis??was it performed??Materials and methods????results???
References
Most of the references (39) have more than 5 years. Please update your list of references.
Author Response
The suggested changes to the Manuscript molecules-923965 entitled “Development of a Third Generation Snack of Rice Starch Enriched with Nopal Flour (Opuntia ficus
indica)”, are provided by the following means. Each of the changes was made and
highlighted in the text with yellow color based on the suggestion of the reviewer. Thank you for your attention. Best regards.
Dr. Tomas Galicia Garcia
Corresponding Author
REVIEWER1
Comments and Suggestions for Authors
The English language must still be revised.
The writing and spelling in English language was checked by a specialist. Corrections made appear throughout the text.
Results and discussion
Line 357- Table 8, lack of fit of the used models???
In the data analysis the lack of fit does not appear. (see file)
Line 484- “a characterization was made in commercial snacks of the brands: "Valentones Barcel
(VB)", "Mimarca (M) wheat flour rinds and "Traditional snack (TS)”??Please do not use the brands’ names and codify each brand by letters or numbers. Change was realized in line 485.
Line 695- which constraints were used for optimization???experimental validation results of the optimized solutions?????how many replicates??
The answers AD, EI, H, L and a* value were used. Below are the graphs used for graphical optimization (see file). Each result is the average of at least three repetitions.
Materials and methods
Line 570- “an oven with air circulation (75 m/s) until”??75 m/s of air speed??not too high??Please check the used air speed. The correction was made in line 569
Conclusion
Line 712- “Third generation snacks made with extrusion modified starch with formulation 1 (5 % NF and 0 % XG) and with formulation 7 (5 % NF and 0 % XG) were the best sensory evaluated, with respect to color, flavor, general acceptability and especially in texture.”???sensory analysis??was it performed??Materials and methods????results??? Correction was made in lines 712-713
References
Most of the references (39) have more than 5 years. Please update your list of references.
The list of references was updated. Corrections made appear throughout the text and references section.

Reviewer 2 Report
I have read the revised content. In view of the modifications, the format has changed a lot and the logical structure of the article is more coherent. recommended publication. But there are still a few problems need to be amend:
- From 321 to 336, make sure it is IE or EI
- In line 326, the coefficient of X1X2 in the formula should be -0.73.
Author Response
The suggested changes to the Manuscript molecules-923965 entitled “Development of a Third Generation Snack of Rice Starch Enriched with Nopal Flour (Opuntia ficus
indica)”, are provided by the following means. Each of the changes was made and
highlighted in the text with green color based on the suggestion of the reviewer.
Thank you for your attention. Best regards.
Dr. Tomas Galicia Garcia
Corresponding Author
REVIEWER 2
Comments and Suggestions for Authors
I have read the revised content. In view of the modifications, the format has changed a lot and the logical structure of the article is more coherent. recommended publication. But there are still a few problems need to be amend:
1. From 321 to 336, make sure it is IE or EI Change was made
2. In line 326, the coefficient of X1X2 in the formula should be -0.73. Change was made
